

# From the cage to the wild: introductions of Psittaciformes to Puerto Rico

Wilfredo Falcón[1,2,3] and Raymond L. Tremblay[1,2]

[1] Department of Biology, University of Puerto Rico at Humacao, Humacao, Puerto Rico, United States of America
[2] Center for Applied Tropical Ecology and Conservation, University of Puerto Rico, Río Piedras, Puerto Rico, United States of America
[3] Bureau of Research and Conservation of Habitats and Biodiversity, Puerto Rico Department of Natural and Environmental Resources, San Juan, Puerto Rico, United States of America

## ABSTRACT

Introduced psittacine birds can become highly invasive. In this study, we assessed invasions of Psittaciformes in Puerto Rico. We reviewed the literature, public databases, citizen science records, and performed in situ population surveys across the island to determine the historical and current status and distribution of psittacine species. We used count data from *Ebird* to determine population trends. For species whose populations were increasing, we modelled their potential distribution using niche modeling techniques. We found 46 Psittaciformes in Puerto Rico, of which 26% are only present as pets, at least 29 species have been reported in the wild, and of those, there is evidence that at least 12 species are breeding. Our results indicate that most introduced species which have been detected as established still persist, although mostly in localized areas and small populations. Clear evidence of invasiveness was found for *Brotogeris versicolurus* and *Myiopsitta monachus*, which have greatly expanded their range in recent years. *Psittacara erythrogenys* and *Eupsittacula canicularis* also showed population increases, although to a lesser degree. The niche models predicted suitable areas for the four species, and also indicate the potential for range expansion. We discuss the factors leading to invasion success, assess the potential impacts, and we discuss possible management strategies and research prospects.

Corresponding author
Wilfredo Falcón,
wfalcon.research@gmail.com

## INTRODUCTION

With the globalization of economies, the rate of species introductions has risen considerably to the extent that the shifts in species distributions and the reorganization of biodiversity are now considered a signature of the Anthropocene (*Seebens et al., 2017*). In parallel, invasive species have gained broad attention from ecologists, government agencies and the public due to the potential and realized negative impacts on economies, human health, native species, and ecosystem services (*Sanders et al., 2003*; *Mooney, 2005*; *Davis, 2009*; *Lockwood, Hoopes & Marchetti, 2013*). One of the factors that most contributed to the establishment of non-native species in recent history has been the pet trade (*Smith et al., 2009*).

Psittaciformes are among the exotics species most commonly sold as pets, with two-thirds of the known species in this group known to occur in the pet trade (parrots, conures and cockatoos) (*Cassey et al., 2004b*). In the 1990's the global pet trade in Psittaciformes generated USD $1.4 billion and was largely supplied by four million wild-caught birds (*Thomsen et al., 1992*). Four general modes of introductions to native habitats have been identified for parrots: releases by traders due to oversupply or legal complications (*Forshaw, 1973*; *Robinson, 2001*), or the accidental or intentional releases by pet owners (*Blackburn, Lockwood & Cassey, 2009*).

Two thirds of successful avian introductions have been on islands (*Blackburn, Lockwood & Cassey, 2009*), however, it should be noted that most unsuccessful introductions have not been documented (*Mori et al., 2014*). In Puerto Rico, many species of Psittaciformes have been imported for sale as pets, especially since the 1950's, and by 2000, eight had likely become established (*Pérez-Rivera, 1985*; *Raffaele, 1989*; *Camacho-Rodríguez, Chabert-Llompart & López-Flores, 1999*; *Oberle, 2000*). As with any exotic, invasive species, local, state, and federal agencies are concerned with the possible effects that these species may have on ecosystem processes and populations of indigenous species. Of particular concern in Puerto Rico is the US Federal Government and World Conservation Union listed, critically endangered species, the endemic Puerto Rican amazon, *Amazona vittata* (*Snyder, Wiley & Kepler, 2007*).

In this study, we review the introduction and persistence of Psittaciformes in Puerto Rico, by evaluating their historic and present distributions. In addition, we assess population trends, and present the predicted distribution of the most successful psittacine species on the island using niche modelling techniques. Finally, we identify possible factors that may have contributed to the successful establishment of Psittaciformes in Puerto Rico and discuss our results in the context of potential impacts, management and prospects.

## MATERIALS AND METHODS

Members of the Psittaciformes, which comprise about 393 species in 92 genera that include macaws, cockatoos, parrots, and conures, are mostly pantropical, although some species inhabit temperate areas in the southern hemisphere (*Juniper & Parr, 1998*; *Forshaw, 2010*). They are considered one of the most endangered groups of birds in the world, and threats include trapping for trade, habitat destruction and hunting (*Snyder et al., 2000*).

### Historical and current status of Psittaciformes

To assess the historical introductions of Psittaciformes to Puerto Rico, and their current status, we surveyed historical reports on the distribution of the species (*Forshaw, 1973*; *Pérez-Rivera & Vélez-Miranda, 1980*; *Pérez-Rivera, 1985*; *Pérez-Rivera et al., 1985*; *Raffaele, 1989*; *Camacho-Rodríguez, Chabert-Llompart & López-Flores, 1999*; *Oberle, 2000*) . We also recorded species occurrences based on observations made during 2013–2017. In addition, we surveyed the *E-bird* (http://ebird.com/) online database, which contain records from amateurs and professional ornithologists.

Well curated open access observational data, such as that available through *Ebird*, provides a base for evidence-based research, conservation and management actions in

a cost-effective way, especially in the absence of scientifically-derived data (*Sullivan et al., 2017*). Still, these data are not free of biases. To collect data, *Ebird* employs relatively unstructured protocols with the aims of attracting a large number of observers (*Hochachka, Fink & Zuckerberg, 2012*). As a result, these protocols do not control for several sources of error during the data collection process, which need to be accounted for during data manipulation and analysis, (e.g., variation in the observation process such as duration of observation periods, distances travelled and time of day, can affect the probability of detecting birds; *Sullivan et al., 2014*; *Sullivan et al., 2017*). After data collection, the results that become part of the data products accessible through *Ebird* are validated through automated and human screening of the observations (*Sullivan et al., 2017*). For example, escaped pets are likely part of the records contained in *Ebird*, which can bias the distribution models (see below). Nonetheless, we assume that sampling effort is equal across years in our estimation of sighting trends, which is not likely the case. This could bias the estimation of population increase based on sighting trends. Therefore, our results must be interpreted within the boundaries of these limitations.

Searches were also conducted on online local birding groups for photographic records using the search terms 'parrot', 'parakeet', 'macaw', 'cockatoo' in English and Spanish. The local groups included: *Aves de Puerto Rico* (https://www.facebook.com/avesdepuertoricoFelPe/), the *Puerto Rico Ornithological Society* (https://www.facebook.com/sociedadornitologicapuertorriquena/), *Bird Photographers of Puerto Rico* (https://www.facebook.com/groups/615958701756859/), and *Biodiversidad de Puerto Rico* (https://www.facebook.com/groups/PRNatural/). This was done between November–December 2017 (see Supplemental Information S1 for the origin of the data). To identify species sold as pets we visited pet stores, mainly in the Metropolitan Area of San Juan, and the pet section of local online classified ads *Clasificados Online* (http://www.clasificadosonline.com/). Moreover, the online databases *Ebird*, *CABI Invasive Species Compendium* (https://www.cabi.org/isc/) and the *Global Invasive Species Database* (http://www.iucngisd.org/gisd/) were used to assess the "invasiveness" of species of Psittaciformes have reached in their non-native range (outside Puerto Rico) using the categorization scheme of *Blackburn et al. (2011*; see Table 1). The latter scheme was also used to classify the invasive status of introduced psittacines in Puerto Rico. We used *Forshaw (2010)* for taxonomical classification and common names of Psittaciformes, the native distribution, and include any classification changes according to *Del Hoyo et al. (2014)*. The IUCN Red List (ver. 3.1; http://www.iucnredlist.org) was used to assess the conservation status for each species, population trends in their native range, and possible threats or reasons for population increase.

## Sighting trends of Psittaciformes in Puerto Rico

To assess the sighting trends of Psittaciformes in Puerto Rico, we used observation count reports from *Ebird* for each species (see Supplemental Information S2 for count data). We calculated the mean number of birds counted per municipality/year (where they have been observed), and then summed the mean number of birds per municipality to obtain

**Table 1** **Introduced Psittaciformes reported in Puerto Rico, their native range, invasiveness and their current status on the island.** We list the conservation status in their native range under the Red List (v. 3.1) as least concern (LC), vulnerable (VU), near threatened (NT), and endangered (EN). 'Invasiveness' represent the invasion stage reported outside of Puerto Rico (invasiveness potential) and 'Status' represent the invasion stage at which a species is at the moment in Puerto Rico, according to the categorization scheme by *Blackburn et al. (2011*; see below for the definitions). 'Pet trade' indicates if the species is known to be currently sold as pet (Y) or unknown (U) in Puerto Rico. Basis of records are from observations by the authors during the surveys, historical records in the literature (prior to 2000), geo-referenced records from online databases, and/or citizen-science records from local birding groups (see Supplemental Information S1 for the origin of the data). Definitions of invasiveness and status are as follow: (A) Not transported beyond limits of native range (B1) Individuals transported beyond limits of native range, and in captivity or quarantine (i.e. individuals provided with conditions suitable for them, but explicit measures of containment are in place) (B2) Individuals transported beyond limits of native range, and in cultivation (i.e. individuals provided with conditions suitable for them but explicit measures to prevent dispersal are limited at best) (B3) Individuals transported beyond limits of native range, and directly released into novel environment (C0) Individuals released into the wild (i.e. outside of captivity or cultivation) in location where introduced, but incapable of surviving for a significant period (C1) Individuals surviving in the wild (i.e. outside of captivity or cultivation) in location where introduced, no reproduction (C2) Individuals surviving in the wild in location where introduced, reproduction occurring, but population not self-sustaining (C3) Individuals surviving in the wild in location where introduced, reproduction occurring, and population self-sustaining (D1) Self-sustaining population in the wild, with individuals surviving a significant distance from the original point of introduction (D2) Self-sustaining population in the wild, with individuals surviving and reproducing a significant distance from the original point of introduction (E) Fully invasive species, with individuals dispersing, surviving and reproducing at multiple sites across a greater or lesser spectrum of habitats and extent of occurrence.

| Species | Common name | Red List | Native range | Invasiveness | Status | Pet trade |
|---|---|---|---|---|---|---|
| *Agapornis fischeri* | Fischer's lovebird | NT | Tanzania | D1–D2 | C0–C2 | Y |
| *Agapornis personatus* | Masked lovebird | LC | Tanzania | D1–D2 | C0–C2 | Y |
| *Agapornis roseicollis* | Peach-faced lovebirds | LC | Southern Africa | C0–C2 | C0–C2 | Y |
| *Amazona aestiva* | Blue-fronted amazon | LC | South America | C0–C2 | C0–C2 | Y |
| *Amazona albifrons* | White-fronted amazon | LC | Central America | C0–C2 | C3 | Y |
| *Amazona amazonica* | Orange-winged amazon | LC | South America | D1–D2 | E | Y |
| *Amazona auropalliata* | Yellow-naped amazon | VU | Central America | C0–C2 | B2 | Y |
| *Amazona leucocephala* | Cuban amazon | NT | Western Caribbean | B2 | C0–C2 | U |
| *Amazona ochrocephala* | Yellow-crowned amazon | LC | South America | D1–D2 | C0–C3 | U |
| *Amazona oratrix* | Yellow-headed amazon | EN | Central America | C0–C2 | C0–C3 | Y |
| *Amazona ventralis* | Hispaniolan amazon | VU | Hispaniola | C0–C2 | C0–C3 | U |
| *Amazona viridigenalis* | Greencheeked amazon | EN | Mexico | C0–C2 | C3 | Y |
| *Anodorhynchus hyacinthinus* | Hyacinth macaw | VU | South America | B2 | B2 | Y |
| *Ara ararauna* | Blue-and-yellow macaw | LC | South America | C0–C2 | E | Y |
| *Ara chloropterus* | Red-and-green macaw | LC | South America | B2 | C0–C2 | Y |
| *Ara macao* | Scarlet macaw | LC | C.-S. America | B2 | C0–C3 | Y |
| *Ara militaris* | Military macaw | VU | South America | C0–C2 | C0–C2 | Y |
| *Aratinga (Nandayus) nenday* | Nanday conure | LC | Northern S. America | D1–D2 | C0–C2 | Y |
| *Aratinga jandaya* | Jandaya conure | LC | Brazil | B2 | B2 | Y |
| *Aratinga solstitialis* | Sun conure | EN | Brazil, Guyana | C0–C2 | B2 | Y |
| *Brotogeris versicolurus* | White-winged parakeet | LC | South America | E | E | Y |
| *Cacatua alba* | White-crested cockatoo | EN | Indonesia | C0–C2 | C0–C2 | Y |
| *Cacatua galerita* | Sulfur-crested cockatoo | LC | Australasia, Indonesia | D1–D2 | C0–C2 | Y |
| *Cacatua goffiniana* | Goffin's corella | NT | Indonesia | D1–D2 | C0–C2 | Y |
| *Cacatua moluccensis* | Salmon-crested cockatoo | VU | Indonesia | B2 | C0–C2 | Y |
| *Cacatua sulfurea* | Yellow-crested cockatoo | CR | Timor-Leste, Indonesia | B2 | B2 | Y |
| *Eupsittula (Aratinga) canicularis* | Orange-fronted conure | LC | Central America | B2 | E | Y |
| *Eupsittula (Aratinga) pertinax* | Brown-throated conure | LC | Aruba, C.-S. America | C0–C2 | C0–C2 | U |

**Table 1** (*continued*)

| Species | Common name | Red List | Native range | Invasiveness | Status | Pet trade |
|---|---|---|---|---|---|---|
| *Forpus passerinus* | Green-rumped parrolet | LC | Northern S. America | D1–D2 | B2 | Y |
| *Melopsittacus undulatus* | Budgerigar | LC | Australia | C0–C2 | C0–C2 | Y |
| *Myiopsitta monachus* | Monk parakeet | LC | South America | E | E | Y |
| *Nymphicus hollandicus* | Cockatiel | LC | Australia | C0–C2 | C0–C2 | Y |
| *Poicephalus senegalus* | Senegal | LC | Africa | B2 | C0–C2 | Y |
| *Psephotus haematonotus* | Red-rumped parrot | LC | Australia | B2 | B2 | Y |
| *Psittacara (Aratinga) chloropterus* | Hispaniolan conure | VU | Hispaniola | C0–C2 | C0–C3 | U |
| *Psittacara (Aratinga) erythrogenys* | Red-masked conure | NT | Ecuador, Peru | D1–D2 | D2 | Y |
| *Psittacara mitratus* | Mitred conure | LC | South America | B2 | C0–C2 | Y |
| *Psittacula krameri* | Roseringed parakeet | LC | Africa and Asia | E | C0–C2 | Y |
| *Psittacus erithacus* | African grey parrot | EN | Africa | C0–C2 | B2 | Y |
| *Psittacus timneh* | Timneh parrot | EN | West Africa | B2 | B2 | Y |
| *Pyrrhura hoffmanni* | Sulphur-winged parakeet | LC | Central America | B2 | C0–C2 | Y |
| *Pyrrhura molinae* | Green-cheeked conure | LC | South America | B2 | C0–C2 | Y |
| *Pyrrhura perlata* | Crimson-bellied conure | VU | South America | B2 | B2 | Y |
| *Pyrrhura roseifrons* | Rose-fronted parakeet | LC | South America | B2 | C0–C2 | U |
| *Thectocercus acuticaudatus* | Blue-crowned parakeet | NA | South America | C0–C2 | C0–C2 | U |
| *Trichoglossus haematodus* | Rainbow lorikeet | LC | Australasia, Indonesia | C0–C2 | B2 | Y |

the island-wide counts per year. Only species with at least 20 records were included in the subsequent analyses.

## Distribution of Psittaciformes in Puerto Rico

We assessed the current distribution of the introduced psittacines by surveying *Ebird* for geo-referenced records (see Supplemental Information S2 for location records). After identifying psittacine species whose sighting trends showed an increase, we employed niche-modeling techniques following the methodology used by *Falcón, Ackerman & Daehler (2012)* and *Falcón et al. (2013)*. Species distribution models were constructed with the Maximum Entropy Method (MaxEnt ver. 3.4.1; *Phillips, Anderson & Schapire, 2006*; *Phillips, Dudík & Schapire, 2017*), which is a learning machine method that uses presence only data in combination with predictive variables to model a species' geographic distribution. Several studies have demonstrated that MaxEnt performs well at predicting species geographic distributions, and it has shown better performance and predictive ability when compared to other niche-modeling techniques (e.g., *Duque-Lazo et al., 2016*; *Bueno et al., 2017*). We used climatic layers obtained from WorldClim (ver. 2) as predictive variables in our model (http://worldclim.org/version2; *Fick & Hijmans, 2017*). These variables are derived from monthly temperature and rainfall values that represent annual trends, seasonality and extreme or limiting environmental factors (*Hijmans et al., 2005*). For our models, we selected temperature seasonality (BIO4), maximum temperature of the warmest month (BIO5), minimum temperature of the coldest month (BIO6), precipitation of the wettest month (BIO13), precipitation of the driest month (BIO14) and precipitation seasonality (BIO15) as climatic variables because they represent the extreme limiting climatic factors and their variation. The calibration area was defined as the smallest

rectangle that encompassed all the location records that we used for the model plus 10 km, and restricted to the subtropics in the Americas (minimum and maximum latitude −30–30°; *Corlett, 2013*). For presence records, we used the geo-referenced and validated locations that we obtained from the different sources listed above, after eliminating duplicates and setting a minimum distance of 1.5 km between occurrence records (to prevent model overfitting due to similar climatic conditions of adjacent points). We randomly selected 20% of the points to test the performance of the model and performed 10 replicates. To evaluate the performance of the model, we used the AUC statistics (area under the receiver operating characteristic curve), which provides a single measure of the model performance (*Phillips, Anderson & Schapire, 2006*). Models that have an excellent predictive performance have AUC values >0.90, 0.80–0.90 are considered to have good predictive performance, while models with AUC values <0.70 are considered poor (*Swets, 1988*; *Manel, Williams & Ormerod, 2001*; *Franklin, 2010*). To determine the presence–absence threshold, we used the Maximum Training Specificity plus Sensitivity threshold, which minimizes the mean of the error rate for positive and negative observations (*Manel, Williams & Ormerod, 2001*; *Freeman & Moisen, 2008*), and performs better than other thresholds in providing accurate presence predictions (*Liu et al., 2005*; *Jiménez-Valverde & Lobo, 2007*; *Freeman & Moisen, 2008*).

### Habitat association of Psittaciformes in Puerto Rico

To characterize the habitat used by non-indigenous psittacines whose populations are increasing, we extracted land cover data from the Puerto Rico Land Cover Map (*Gould et al., 2007*) using the occurrence locations obtained from *Ebird*. We restricted this analysis to occurrence records from 2000–2017 and assume no land cover changes after the map was developed (mostly from satellite images from 2001–2003). We simplified the land cover categorization into 'urban' 'forests', 'pastures', 'wetlands' and 'other' habitats (see Supplemental Information S3 for original and simplified classifications).

### Statistical analyses and data visualization

We performed all data pre-processing, and obtained summary statistics and visualizations using R ver. 3.3.3 (*R Core Team, 2017*), and packages 'ggplot2' (*Wickham, 2016*), 'raster' (*Hijmans et al., 2017*), 'zoo' (*Zeileis & Grothendieck, 2005*), 'plyr' (*Wickham, 2011*), and the output from MaxEnt. We mapped occurrence records and visualized distribution results using QGIS ver. 2.18.14-Las Palmas de G. C. (*QGIS Development Team, 2017*). To test whether the observed habitat used by the species follows the proportion of the available habitat, we compared the observed habitat used by each species to the available habitats throughout Puerto Rico (extracted from the Land Cover Map) using Chi-square goodness-of-fit analysis (*Neu, Byers & Peek, 1974*) and computed the *p*-values for a Monte Carlo test (*Hope, 1968*).

## RESULTS

### Historical and current status of Psittaciformes

We found historical records for 18 species of Psittaciformes reported by the year 2000, with eight of those breeding. At least 46 psittacine species are now present on the island

(Table 1; Fig. 1), of which 24% are only found in the pet trade, 48% have been observed in the wild (present), but not known to be breeding (established), and 28% are established (naturalized) and known to have bred or are currently breeding (Table 1). At least 85% of the species are currently available for sale in the pet trade. Of the 46 species of Psittaciformes found in Puerto Rico, at least 63% have been reported in the wild elsewhere (but it is unknown whether they are breeding in these new environments), and 26% are considered as established or invasive (i.e., breeding outside their native range and expanding their range).

## Sighting trends of Psittaciformes in Puerto Rico

We found sufficient reports to calculate the island-wide sighting trends for 10 species of Psittaciformes in Puerto Rico (Fig. 2). Four species exhibited population increase, three species showed stable populations, and three species exhibited population decrease. For the rest of the species, the count numbers were too low and/or temporal resolution was too short to calculate trends. Of the species with growing populations, the white-winged parakeet (*Brotogeris versicolurus*) showed the largest population increase, followed by the monk parakeet (*Myiopsitta monachus)*, the red-masked conure (*Psittacara (Aratinga) erythrogenys*) and the orange-fronted parakeet (*Eupsittula canicularis;* Fig. 2). The orange winged amazon (*Amazona amazonica*) showed a population increase and later stabilized, while the white-crested cockatoo (*Cacatua alba*) showed a stable population trend, albeit with low numbers. The blue-and-yellow macaw (*Ara aranaura*) exhibited a decrease from 1996 to 2002 (with up to 22 individuals reported) in the Metropolitan Area of San Juan, and later recovered and stabilized with about 14 individuals. The green-cheeked amazon (*Amazona viridigenalis*) showed a steep decrease after reaching a total of 53 individuals, with low counts after 2006. Similar trends were observed in the white-fronted amazon (*A. albifrons*), but in much lower numbers. Finally, the rose-ringed parakeet (*Psittacula krameri*) showed a sustained decrease from a maximum of 12 individuals which was reported in 2011.

The orange-winged amazons exhibited an increase in sightings since the 1990's, and by 2014, the population seemed to be stabilized. The maximum number of individuals reported is 39, but there is a population roosting in the municipality of Morovis with over 100 individuals (not included in the count data; J Salgado Vélez, pers. comm., 2018). Moreover, the orange-winged amazons are the most widespread of the amazon parrots in Puerto Rico. The blue-and-yellow macaws apparently experienced a population decline in the Metropolitan Area of San Juan, and later increased and stabilized. Furthermore, a population of at least 15 individuals is breeding in the municipality of Orocovis, ~29 km away from the other population (not included in count data), and some have been sighted in Cabo Rojo (southwest Puerto Rico). The sighting trends of the white-crested cockatoo indicate that the population is small but stable (2–15 individuals) and localized in the adjacent municipalities of Bayamón and Guaynabo in the Metropolitan Area. Other species exhibited sighting declines after an initial increase. The green-fronted amazon has two reported populations; one in the municipality of Mayagüez (west) with up to 10 individuals recorded, and one in the municipality of Salinas (southeast) with up to

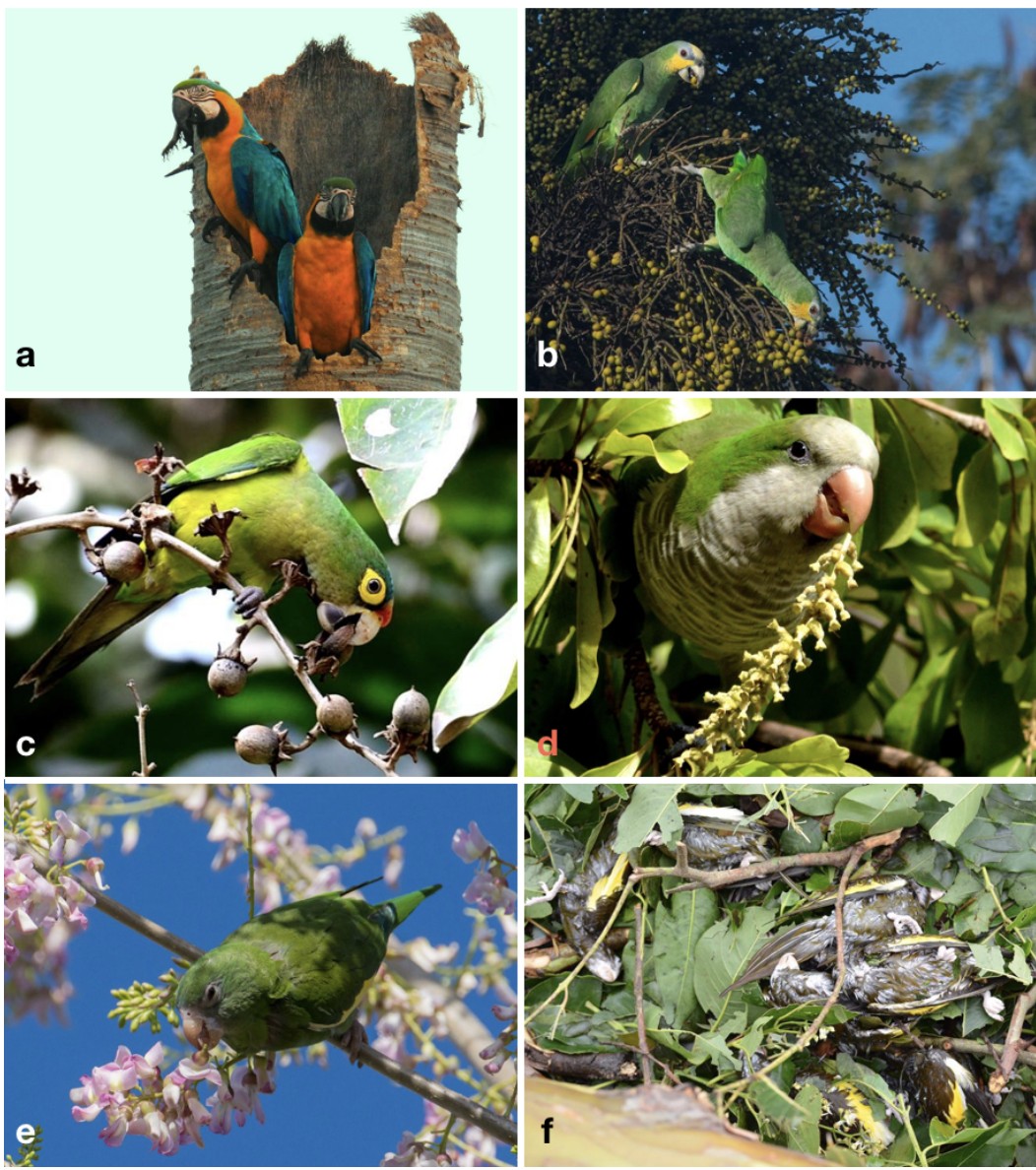

**Figure 1** **Some of the species of Psittaciformes that occur in the wild in Puerto Rico, and Hurricane Maria-related mortality.** A pair of blue-and-yellow macaws (*Ara ararauna*) in their nest on *Roystonea borinquena* (Aracaceae; a), an orange-winged amazon (*Amazona amazonica*) eating palm fruits (Aracaceae; b), an orange-fronted parakeet (*Eupsittula canicularis*) foraging on seeds of *Lagerstroemia speciosa* (Lythraceae; c), a monk parakeet (*Myiopsitta monachus*) eating the flower buds of *Bucida buceras* (Combretaceae; d), a white-winged parakeet (*Brotogeris versicolurus*) eating flower buds (Fabaceae; e), and six out of dozens of white-winged parakeets that died during Hurricane Mari'a in 2017 (f). Photo credits: Yoly Pereira (A), Julio Salgado (B, E), Pedro Santana (C), Sonia Longoria (D), Dinath Figueroa (F).

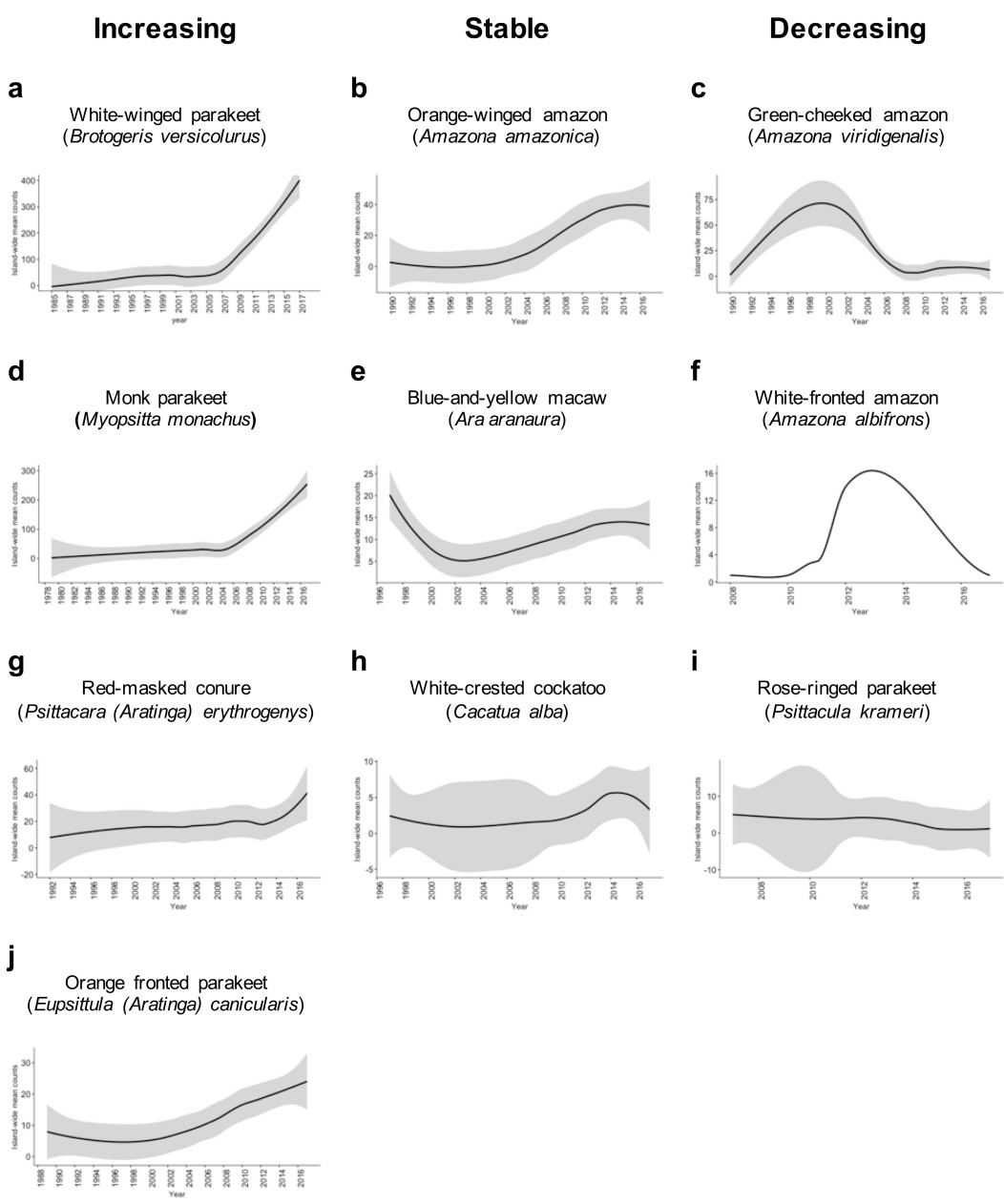

**Figure 2** **Sighting trends of different species of Psittaciformes in Puerto Rico showing population increase (A, D, G, J), stable populations (B, E, H) and population decrease (C, F, I).** Island-wide sighting trends were calculated as the sum of the mean number of birds counted per year/municipality (data from Ebird). Grey shading indicates the 95% CI based on the local weighted scatterplot smoothing (loess).

30 individuals recorded. In the case of the white-fronted amazon, which is restricted to Mayagüez, it exhibited a reduction in sightings, from up to 11 individuals in 2011, to 1–2 individuals in recent years. The rose-ringed parakeet is currently uncommon, but in 2012 at least 12 individuals were sighted in Aguadilla (northwest Puerto Rico).

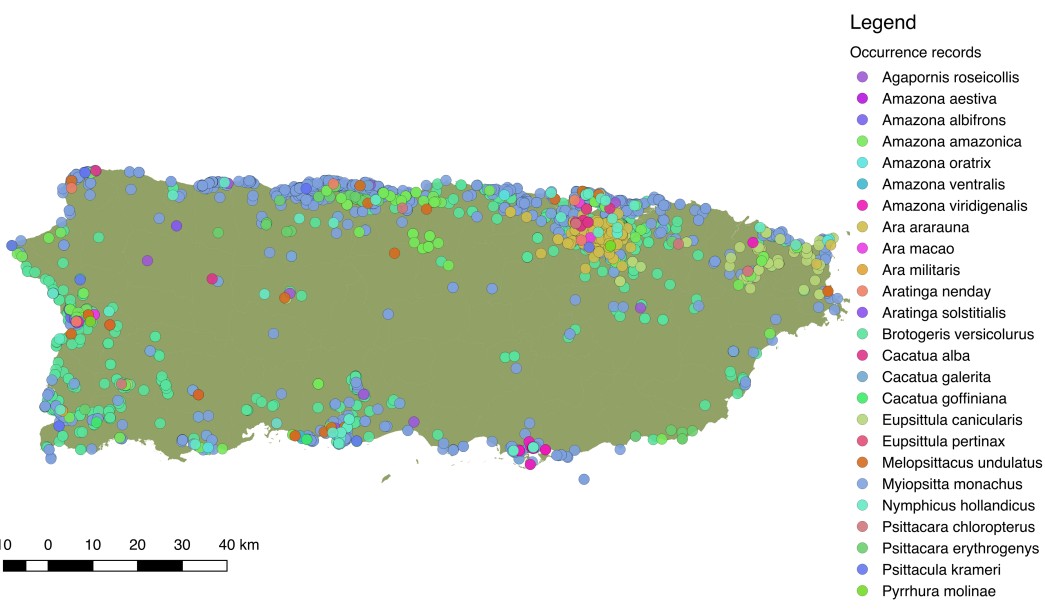

**Figure 3** **Distribution of 25 species of Psittaciformes in Puerto Rico, depicted by the different colors.** Records originated from observations made by the authors, online databases, and reports from local birding groups (see 'Methods').

## Habitat association of Psittaciformes in Puerto Rico

All four species of Psittaciformes whose populations were increasing (white-winged parakeets, monk parakeets, red-masked conure and the orange-fronted parakeet) exhibited similar habitat use; 66% of the occurrences were located in urban areas (33% in high density and 33% in low density urban areas) while 33% of the other occurrences were located in forested areas near urban areas. The observed habitat used differed from the expected habitat available for all four species ($p <$ 0.001 in all cases).

## Distribution of Psittaciformes in Puerto Rico

Overall, we obtained 6,905 locality records for 26 species of Psittaciformes in the wild that spanned from 1960 to 2017 (in *Ebird*; see Supplemental Information S1 for detailed species-specific results and S2 for location records). Moreover, we obtained 279 sighting records for 10 species from local groups (see Supplemental Information S4 for links to species-specific reports). Sightings and reports of Psittaciformes in Puerto Rico practically cover the whole island and are especially dense in coastal and highly (human) populated areas (Fig. 3), but the geographic extent of the distribution varies by species (see Supplemental Information S5 for the distribution of locations for each species reported in the island).

More specifically, we obtained 2,520 occurrences for the white-winged parakeet. Historic reports on the distribution of *B. versicolurus* show that parakeets were present in Luquillo (east) in small numbers by the 1960's (Kepler; in *Bond, 1971*). Moreover, a small population was found breeding in the municipality of Naguabo (east) and about 360 individuals were reported in Guaynabo (San Juan metro area) by 1985, where the San Patricio population is presently located (*Pérez-Rivera et al., 1985*). Furthermore, the only other historic records are

from the population in San Germán (southwest), which was estimated at 800 individuals by 1995 (*Camacho-Rodríguez, Chabert-Llompart & López-Flores, 1999*). Since then, the parakeet populations have expanded significantly throughout the island, especially to coastal and urban/suburban areas, but occasionally they have been observed in the central mountainous regions (Fig. 4A).

To predict the potential distribution of the psittacine species whose populations are increasing in Puerto Rico, we obtained 106,493 occurrence records from their native and invasive range (including Puerto Rico; see Supplemental Information S2). Our models performed good to excellent, with test AUC values ranging from 0.82–0.94 (Table 2), and the presence records for the four species are within the predicted distribution of their native ranges, indicating a good model fit. Furthermore, the models predicted suitable areas for all four species in Puerto Rico (Fig. 4).

The predicted distribution for the white-winged parakeet in Puerto Rico showed the highest suitable areas in the north-central part of the island, but also included other coastal areas. Only in the central and central-west part of the island, where the Central Cordillera occurs, did MaxEnt predict unsuitable areas for the species. In general, the white-winged parakeet occupies virtually all areas with the highest suitability. Similar areas were predicted with suitable climatic conditions for the monk parakeet, although there are areas in the west of the island predicted as unsuitable where the monk parakeet has been reported. The models also predicted suitable areas for the red-masked conure and the orange-fronted parakeet, some of which they occupy. Both species have lower predicted climatic suitability than white-winged and monk parakeets, and the orange-fronted parakeet has a smaller predicted suitable area than the other species. In general, our models predicted suitable areas outside the current range of all four species, indicating the possibility for further range expansion.

## DISCUSSION

In this study, we assessed the status of non-native Psittaciformes in Puerto Rico. All have been introduced via the pet trade, and nearly half of the 46 psittacine species present on the island have been observed in the wild. Historical records indicate that once established and breeding, most are persistent.

Observations on flock size and sighting trends of Psittaciformes in Puerto Rico indicate that the different species are experiencing different dynamics. White-winged and monk parakeets are the only two species that appear to be growing exponentially. Moreover, for the white-winged parakeet, the island-wide sum of the mean counts per municipality shows a substantial increase of birds across Puerto Rico, indicating range expansion and exhibiting a lag-phase followed by exponential growth. This is consistent with the patterns of increase and population growth rates in two of the largest populations of white-winged parakeets in Puerto Rico (*Falcón & Tremblay, 2018*). Similarly, records for the monk parakeets started in 1979 throughout coastal and densely populated (humans) areas (mainly San Juan and Ponce), and by 1987, flocks of up to 30 individuals were observed. By the late 1990's sightings outside these two municipalities started and became more common throughout the island as time passed probably as a result of population growth and range expansion.

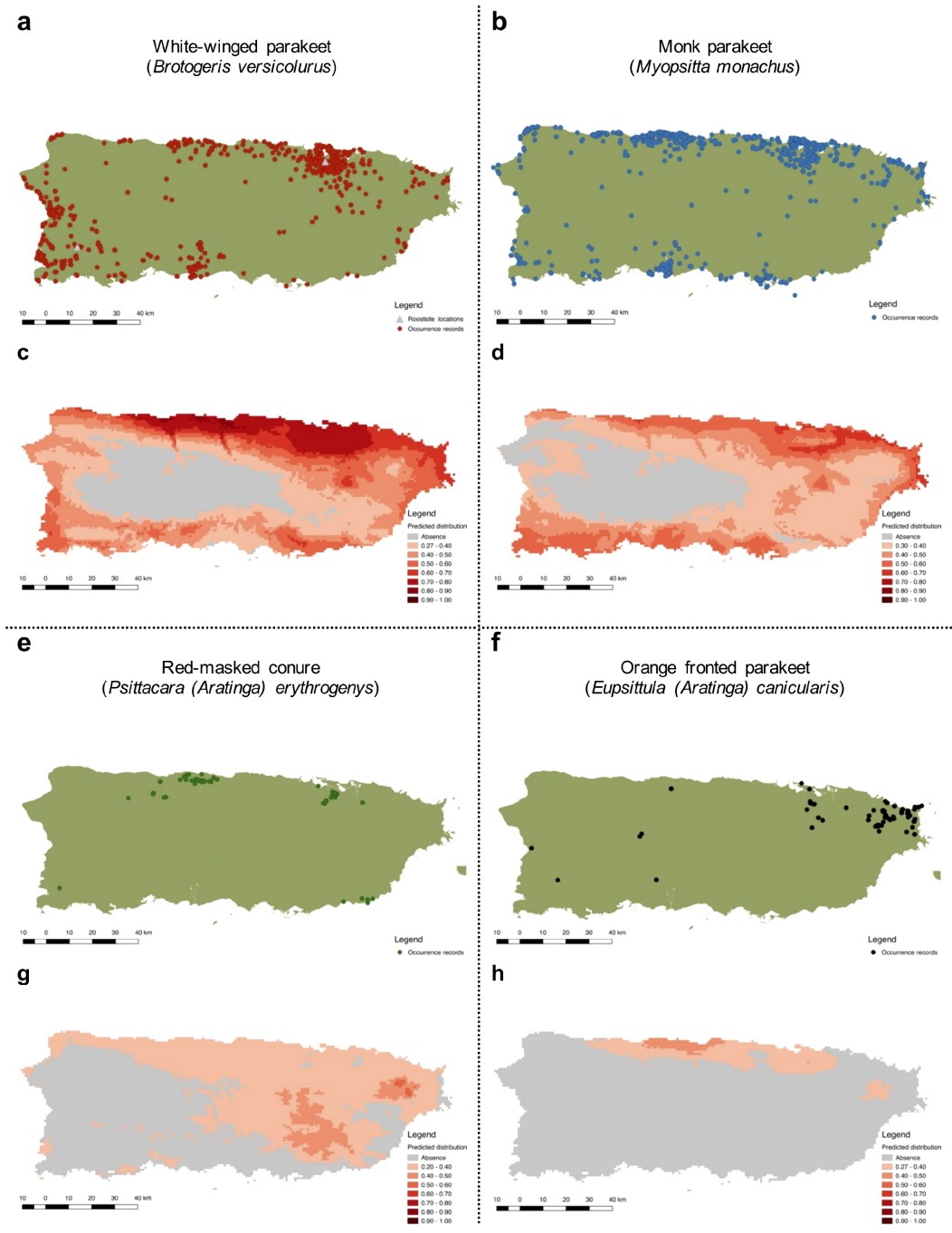

**Figure 4 Current distribution of Psittaciformes in Puerto Rico whose populations are increasing (A, B, E, F) and the predicted distribution of the species based on the maximum entropy model (MaxEnt: C, D, G, H).** Warmer colors depict higher suitability.

**Table 2 Occurrence locations and MaxEnt summary statistics for predicting the distribution of four species of Psittaciformes in Puerto Rico.** Total occurrence locations used for the model (Occ. loc.), and the subset of occurrence locations for Puerto Rico given in parentheses. Model performance for predicting the distribution of four species of Psittaciformes in Puerto Rico given by the training (80% occurrence points) and test (20% occurrence points) Area under the ROC Curve (AUC), and the Maximum training Sensitivity plus Specificity presence-absence threshold (MS +ST). Values given are the mean (±SD) based on ten replicates.

| Species | Occ. loc. | Training AUC | Test AUC | MS+ST |
|---|---|---|---|---|
| *Brotogeris versicolurus* | 1,119 (557) | 0.91 (±0.001) | 0.92 (±0.006) | 0.27 (±0.024) |
| *Myopsitta monachus* | 15,985 (707) | 0.82 (±0.001) | 0.82 (±0.004) | 0.30 (±0.031) |
| *Psittacara erythrogenys* | 1,392 (42) | 0.94 (±0.002) | 0.94 (±0.003) | 0.20 (±0.036) |
| *Eupsittula canicularis* | 3,707 (55) | 0.84 (±0.001) | 0.84 (±0.005) | 0.27 (±0.014) |

In general, invasive Psittaciformes are strongly associated with urban areas (e.g., *Avery & Shiels, 2017*) and those in Puerto Rico are no exception. These birds are most commonly sighted along coastal regions of the island, particularly areas with the highest human population densities, as in the metropolitan area of San Juan. Nevertheless, the location and the extent of the geographic distribution of Psittaciformes in Puerto Rico varies among species. Psittacine species whose populations are increasing in Puerto Rico occur in similar habitats and more frequently in urban rather than forested areas. Yet even those wooded areas are close to urban centers where they are known as 'novel' forests for their mixture of native and invasive species (*Lugo, 2004*; *Gould et al., 2007*).

Populations of the most recorded and widespread species, monk and white-winged parakeets, are increasing, and their current distributions coincide with areas predicted to be climatically suitable. Moreover, suitable areas are present outside where these different species occur, indicating the potential for continued range expansion.

When modelling the potential distribution of invasive species, especially when performing risk analyses, a more complete envelope of conditions (climatic in our case) in which the species can survive and reproduce can be attained by including a combination of presence records from both the native and invasive range (e.g., *Steiner et al., 2008*; *Falcón, Ackerman & Daehler, 2012*; *Falcón et al., 2013*). Despite doing so in our analyses, and having models with good predictive scores (high AUC values), some presence records fell out of the predicted distributions, even for the most abundant and widespread species (e.g., locations in the central-western parts of the island). Several factors may have contributed to this. First, some records may reflect locations where the species are only transient or represent locations of recently escaped pets. Secondly, inherent uncertainties and error are associated with the models; AUC values, while good to excellent (0.82–0.94), were not perfect. Nevertheless, the models predict habitat suitability for all locations where we know that the different species reside and reproduce (based on our surveys).

Perhaps the most important factor promoting the success of Psittaciformes outside their native range is the sheer number of individuals that were and are available through the pet trade. Invasion success for exotic bird species is positively influenced by the number of individuals available on the market (*Carrete & Tella, 2008*). Species most commonly found in captivity and those that are inexpensive, have a higher probability of being introduced

into the wild (*Robinson, 2001*; *Cassey et al., 2004b*; *Blackburn, Lockwood & Cassey, 2009*). In general, among bird species, Psittaciformes have a high probability of transport and introduction outside their native range (*Lockwood, 1999*; *Lockwood, Brooks & McKinney, 2000*; *Blackburn & Duncan, 2001*; *Duncan, Blackburn & Cassey, 2006*), being traded up to 14 times more often than other avian orders (*Bush, Baker & MacDonald, 2014*). Moreover, parrots in general have a wide diet breath, and research suggests that diet breath and migratory tendencies can explain the success of established exotic populations of parrots (*Cassey et al., 2004a*). Consequently, 10–16% of all parrot species have established exotic populations around the world (*Cassey et al., 2004b*; *Menchetti & Mori, 2014*). Parrots have been and continue to be popular pets in Puerto Rico, so it is likely that propagule pressure has played a role in the introduction of psittacines on the island. Moreover, their wide diet breath and their affinity for urban habitats may have helped them establish wild populations. This seems to be the case with the white-winged parakeets in Puerto Rico, which remain highly traded on the island (*Falcón & Tremblay, 2018*).

As with any introduced invasive species, there are concerns over the possible negative impacts that exotic parrots may have on the economy, ecosystem services, and populations of native species (*Pérez-Rivera & Vélez-Miranda, 1980*; *Menchetti & Mori, 2014*). These include damage to crops, damage to the electrical infrastructure, transmission of diseases, competition, and hybridization. However, other than warnings about potential impacts, no studies have reported negative impacts by Psittaciformes on ecosystem functions, or specific species in Puerto Rico. Damage to crops is perhaps the biggest negative impact that parrots cause in both their native and introduced range, with substantial economic losses (e.g., *González, 2003*). However, parrots in Puerto Rico seem to be associated with urban areas and heavily rely on food resources in these areas as well as from nearby novel forests, and we found no reports on crop damage. In Florida (USA), monk parakeets have been reported to cause electric shortages while building nests in electric towers, costing as much as USD 585,000 for repairs (*Avery et al., 2006*), but there is no evidence of that occurring in Puerto Rico. Moreover, wild and captive Psittaciformes are known vectors of avian and diseases and parasites, and also human diseases (*Clark, Hume & Hayes, 1988*; *Orosz et al., 1992*; *Magnino et al., 1996*; *Mase et al., 2001*; *Azevedo, 2014*; *Done & Tamura, 2014*; *Briceno et al., 2017*). We found an unidentified species of mite (probably *Pararalichus*) on white-winged parakeets. *Pararalichus* mites have been reported on the white-winged parakeet in its native range, along with *Aralichus cribiformes* and *Echinofemur* sp. and *Rhytidelasma* sp. (*Atyeo, 1989*). In most cases, transfer of mites occurs by physical contact between conspecifics, but there are cases of inter-species transfer (*Dabert & Mironov, 1999*; *Hernandes, Valim & Pedroso, 2016*). Although there are concerns that mites may be vectors for diseases among parrots, most feather mites are considered ecto-commensals and feed on the oils produced by the birds (*Blanco et al., 2001*). Competition for nesting cavities is a major cause of concern because there are native species which depend on this scarce resource, and psittacine secondary cavity nesters can be aggressive competitors (*Snyder, Wiley & Kepler, 2007*; *Strubbe & Matthysen, 2009a*; *Orchan et al., 2012*; *Mori et al., 2017*). Currently, populations of non-indigenous cavity nesters (e.g., *Amazona* spp. and rose-ringed parakeets) are relatively low, whereas the most successful psittacine species in

Puerto Rico build their own nests. Finally, hybridization with congenerics is considered a latent threat to the endemic and endangered Puerto Rican amazon. For example, *Amazona oratrix* and *A. aestiva* are known to hybridize in sympatric regions of their introduced ranges (*Martens, Hoppe & Woog, 2013*). Based on the distribution of non-indigenous *Amazona* spp. in Puerto Rico, hybridization with the Puerto Rican amazon is unlikely at the moment. Nonetheless, it is a future possibility as both endemic and introduced populations continue to grow and expand their ranges. Thus, we have not yet uncovered any negative impacts by psitticines on the island, but the potential exists, which merits monitoring.

Introduced psittacines may actually have positive impacts by filling niches once occupied by Puerto Rico's indigenous parrots, which were much more common in the past and inhabited many of the areas now occupied by exotic species. In fact, the extinct Puerto Rican parakeet (*Psittacara maugei*) was so abundant and widespread that it was, in part, hunted down to extinction because of the damage it caused to the agricultural sector (*Olson, 2015*), as was the fate of the Carolina parakeet, *Corunopsis carolinensis* (*Saikku, 1990*).

Of course, pre-colombian habitats of the Puerto Rican amazon have since been heavily modified by human activities. The forests that now exist are considered 'novel' communities (*Lugo, 2004*), secondary forests with a mix of native and introduced species of plants and animals. Similar to the island's exotic finches (*Rafaelle, 1989*), non-native psittacine species are mainly occupying communities which did not exist in the recent past. It remains to be seen whether or not *A. vittata* will find these communities suitable.

As in other psittacine species, perhaps one of the most important functions performed by indigenous parrots in Puerto Rico were seed predation and seed dispersal. Some parrots are known to be both seed predators and dispersers (*Norconk, Grafton & Conklin Brittain, 1998*; *Francisco, Lunardi & Galetti, 2002*; *Blanco et al., 2015*; *Blanco et al., 2016*; *Blanco, Hiraldo & Tella, 2017*), and both seed predation and seed dispersal have important implications for ecosystems dynamics worldwide, and help regulate plant recruitment, competition, and population structure (*Howe & Smallwood, 1982*; *Hamrick, Murawski & Nason, 1993*; *Nathan & Muller-Landau, 2000*). Unlike other birds, parrots have specialized bills that allows them to access resources, such as hard seeds, that are often not available to other animals, and they often destroy seeds in the wild. For example, the white-winged parakeet is a seed predator of the Panama tree (*Sterculia apetala*), an introduced species in Puerto Rico. But they also eat seeds of native species such as the pink trumpet tree (*Tabebuia heterophylla*). Likewise, blue-and-yellow macaws consume seeds of mahogany, *Swietenia* spp., which are exotic trees whose seeds are too large for other species of birds. On the other hand, as generalist frugivores, parrots also act as seed dispersers via endozoochory (*Blanco et al., 2016*). For example, white-winged parakeets are known to disperse *Ficus* spp., which have fruits containing numerous small, hard-seeds (W Falcón, pers. obs., 2008). Little is known of frugivory and seed predation by parrots in their native ranges, and we are unaware of any such studies on introduced psittacines. Future research should focus on the ecological role of introduced psittacines to assess whether or not impacts occur, positive or negative.

Currently, no species of Psittaciformes is considered illegal in Puerto Rico by the Puerto Rico Department of Natural and Environmental Resources (DNER: *Departamento de Recursos Naturales y Ambientales de Puerto Rico, 2003*). Trapping of exotic birds established in Puerto Rico is allowed by the DNER for exportation only, and the sale in local markets is prohibited. The most commonly trapped bird was the white-winged parakeet; however, the number of individuals trapped is rarely reported and no information of the exportation process is revealed. We recommend that protocols should be modified to obtain such information which would inform management strategies.

Prevention can be the most cost-effective way of dealing with invasive species. Although selling some of the wild trapped species of parrots in the local pet market is illegal, it is common to find people selling them, especially on the internet. Particularly common are budgerigars (*Melopsittacus undulatus*), lovebirds (*Agapornis* spp.), white-winged parakeets and monk parakeets. The rose-ringed parakeet is gaining popularity and it is relatively easy to acquire them. It is highly invasive elsewhere, where has caused negative ecological impacts (*Butler, 2003*; *Strubbe & Matthysen, 2009a*; *Strubbe & Matthysen, 2009b*; *Kumschick & Nentwig, 2010*; *Newson et al., 2011*; *Sa et al., 2014*; *Le Louarn et al., 2016*). Therefore, special attention should be given to this species, and others that have invasiveness potential. We recommend that management agencies prohibit and/or limit the sale and possession of psittacine species that are prone to establish outside their native range. Propagule pressure may be reduced by focusing on the trade of Psittaciformes (and other species). Specifically, we proposed that DNER revise and update Article 7 in the "New Regulation to Govern the Conservation and Management of Wildlife, Exotic Species and Hunting in the Commonwealth of Puerto Rico", which deals with exotic species (*Departamento de Recursos Naturales y Ambientales de Puerto Rico, 2003*).

Several strategies for the control and management of established psittacines are available should they become necessary and/or desirable. Public education programs could be implemented to illustrate the importance of breeding controls and the potential negative effects of releasing exotic animals into the wild. When direct management is necessary, there are three options: (1) trapping and exporting birds, (2) birth control chemosterilants such as Diazacon$^{TM}$ (*Yoder et al., 2007*; *Avery, Yoder & Tillman, 2008*; *Lambert et al., 2010*), and (3) culling (lethal). Control efforts, especially lethal ones, may be hindered by the public, who usually protest these actions as parrots have a high aesthetic value (*Avery & Tillman, 2005*).

It is worth mentioning that catastrophic events may severely alter population sizes of both native and invasive species. In late 2017, Hurricane María, a category four hurricane, drastically altered the entire island of Puerto Rico, and undoubtedly caused negative impacts on the parrot populations in the island. Dozens of dead white-winged parakeets were observed around a recent roost located in Río Piedras (municipality of San Juan; Fig. 1F), and at least one blue-and-yellow macaw was found dead in Guaynabo (part of the Greater San Juan metropolitan area). Despite this, numerous flocks of parakeets and at least eight macaws were seen after the hurricane, so a proportion of individuals of these species survived. Even the endemic Puerto Rican amazon suffered very high hurricane-related losses in the east of the island (R Valentín, pers. comm., 2018). Another negative effect

as a result of the hurricane was the lack of food resources due to the massive exfoliation of food plants. For example, blue-and-yellow macaws, which usually forage high on trees and palms, were observed eating flowers on shrubs as low as 1.5 m from the ground due to the lack of food. A follow-up study on the status of Psittaciformes after the hurricane is recommended, as it is possible that the negative effects may result in the extirpation of some of the species, especially those with small population sizes.

Another aspect to consider is that many species of Psittaciformes found in the wild in Puerto Rico are vulnerable or endangered in their native range, and introduced populations provide the opportunity to conduct experiments and/or to explore management techniques that otherwise would be impossible to perform in their native habitat, both aspects which may aid in the conservation of Psittaciformes in their native habitats around the world. For example, studying the resources used and ecological functions performed by non-native psittacine species in the novel forests of Puerto Rico may help managers understand how the Puerto Rican amazon will react to such novel habitats and pressures, which is important when considering the level of habitat fragmentation on the island as well as the potential for population increase and expansion by the endemic parrot.

## CONCLUSIONS

Our study shows that most Psittaciformes introduced to Puerto Rico through the pet trade are still present and persist in the wild. Moreover, most of them, especially those whose populations are increasing, occur in urban habitats and nearby novel forests. To our knowledge, Puerto Rico has the highest number of wild observed and established exotic parrot species in the world based on surveys in other places where parrot populations occur (e.g., *Runde, Pitt & Foster, 2007*; *Mori et al., 2013*; *Symes, 2014*). Finally, based on population sizes and geographical range within the island, white-winged parakeets and monk parakeets are the most successful Psittaciformes in Puerto Rico.

## ACKNOWLEDGEMENTS

We thank Rafael D. Rodríguez, Linda Ortíz and Rebecca Hernández for their help in the field; Marilyn Colón, Alberto Mercado, Ricardo Valentín (all DNER), and Julio A. Salgado (Puerto Rico Ornithological Society) for providing information on native and invasive psittacines of Puerto Rico. We also thank Noramil Herrera, and Alberto Mercado (DNER) for helping in the search of literature resources. Julio A. Salgado also provided assistance in sharing the online form for reporting Psittaciformes in Puerto Rico among different local groups. We also thank Julio Salgado, Yoly Pereira, Pedro Santana, Sonia Longoria, and Dinath Figueroa for permission to use their photographs. Finally, we thank Dr. James D. Ackerman (University of Puerto Rico), Dr. Patricia Gandini (Universidad Nacional de la Patagonia Austral; PeerJ Academic Editor), and reviewers, including Dr. Donald Brightsmith (Texas A&M University), Dr. Emiliano Mori (Università degli Studi di Siena), and an anonymous reviewer, for their insightful comments and suggestions on the manuscript.

### Funding

The PR-Louis Stokes Alliance for Minority Participation Bridge to the Doctorate Fellowship (HRD-0601843), the Ronald E. McNair Program of the University of Puerto Rico at Humacao (P217A070213), and the Center for Applied Tropical Ecology and Conservation (HRD-0734826) provided funding to Wilfredo Falcón in partial support of this project. The funders had no role in study design, data collection and analysis, decision to publish, or preparation of the manuscript.

### Grant Disclosures

The following grant information was disclosed by the authors:
PR-Louis Stokes Alliance for Minority Participation Bridge to the Doctorate Fellowship: HRD-0601843.
Ronald E. McNair Program of the University of Puerto Rico at Humacao: P217A070213.
Center for Applied Tropical Ecology and Conservation: HRD-0734826.

### Competing Interests

The authors declare there are no competing interests.

### Author Contributions

- Wilfredo Falcón analyzed the data, contributed reagents/materials/analysis tools, prepared figures and/or tables, authored or reviewed drafts of the paper, approved the final draft, performed literature review and surveys.
- Raymond L. Tremblay contributed reagents/materials/analysis tools, authored or reviewed drafts of the paper, approved the final draft, performed surveys.

### Data Availability

The raw data are provided in a Supplemental File.

### Supplemental Information

Supplemental information for this article can be found online at http://dx.doi.org/10.7717/peerj.5669#supplemental-information.

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
