# Peer review of "From the cage to the wild: introductions of Psittaciformes to Puerto Rico"

_PeerJ, doi:10.7717/peerj.5669_

## Round 0.1 · original submission · Major Revisions

Your paper needs major revision to be considered for publication in Peerj. Both reviewers considered it has to be shortened.

You will have to address profound changes in your work such as

1) Display separate native and introduced maps. This will facilitate the interpretation of the readers
2)clearly separate the habitat use of native vs. introduced
3) Discuss the scope of the use of eBird data in general
4) Check writing and other major and minor suggestions reviewed by reviewers

Reviewer 1 ·

Basic reporting

The manuscript is poorly written and is much too long for the research findings. It is receptive in places, and extremely "wordy" and contains extraneous information. It could be reduced by half without loss of important content. Suggested improvements are contained in review sent to the authors.

Experimental design

Use of Max Ent model is appropriate for predicting suitable habitat for the exotic parrots and the census methods used for counting roosting parakeets is appropriate. Remainder of ms. base on natural history observations and anecdotes.

Validity of the findings

Results of the niche model seem reasonable, but authors do not provide adequate discussion of model results especially where observations do not fit model predictions. Conclusions includes some stuff unrelated to their research findings.

Additional comments

This manuscript makes a valuable contribution to our understanding of introduced psittacines on the island of Puerto Rico, by summarizing current status of the introduced species and by using niche modeling to indicate which species have occupied their potential available niches. This is of value for understanding how introduced species adapt to their new environments. The niche modeling predicts that some species have colonized much of the potential available habitat on the island or for some species have much more habitat available for colonization. In the case of the monk parakeet, some sight records have been recorded it in habitat, which was predicted to be unsuitable by the model results. The expansion of the range of the white-winged parakeet recorded in the study is also supported by increased counts of the parakeets at roosts, which would contribute to the range expansion. The niche modeling and population counts are new contribution to our undersanding of exotic species on Puerto Rico. However, the manuscript is poorly written – overly “wordy”, includes extraneous details (not clear how the parasitic mite information is relevant to range expansion), and very repetitive (especially in the Discussion section). The authors use common and scientific names interchangeably in an inconsistent manner throughout the ms. The diet information on the white-winged parakeet is poorly integrated in the discussion of the parakeets population growth and range expansion on the island. Furthermore, the Discussion section includes numerous anecdotes and much speculation, which is unwarranted. The Conclusion section includes discussion of issues, which were not related to their research findings. In addition, considerable space is used to review issues and recommendations related to management and control of exotics (from previous studies and not from their own research findings), rather than focusing on the research findings. The discussion section could be shortened by eliminating the anecdotal material on the various exotic psittacines and by providing an appendix with species accounts summarizing the current status of various exotic psittacines. Overall, the results of this study contribute to our understanding of exotic species and their status on Puerto Rico, but of less interest to those concerned with general topics related to naturalization of non-native species. The manuscript could be shortened by half without loss of critical content. The manuscript requires substantial revision and elimination of extraneous material before it is publishable. My specific comments and suggestions follow below:

Line 88 – “many species have been imported to be sold as pets in PR” – Change to: “many species have been imported for sale as pets…..”

Line 94 – “having gone extinct” Change to: “are now extinct”.

Line 97 - “managing agncies” – Change to: managers want

Line 102 – “alerted” Change to: warned

Line 103 – “there is yet no evidence”. Change to: evidence of crop damage yet.

Line 109 – Change to:.... negative effects that the parrot might have on native biota and agricultural sector if the parakeet kept expanding its range.

Line 111 – Change to: In this study we review the introduction and persistence of Psitta…. by evaluating their historic and present distribution. In addition, we assess the invasive ecology of the white-winged parakeet in more detail, including aspects of its natural history, estimated population size and growth rates and predicted distribution based on niche modeling techniques.

Line 115 – Change to: “Finally, we identify possible factors that may have contributed to the successful establishment of Psitt in PR…..”

Line 122 to 126. Eliminate this section.

Line 131—Change to: We also recorded species occurrences based on observations made during 2013-17.

Line 133 – Change to: “which contain records from amateurs and professional ornithologists. Online searches were also conducted for photographic records from local groups including: “

Line 145-145 – Modify to ….”were used to assess the “invasiveness” of various parrot species in non-native locales (outside of Puerto Rico) using the invasive categorization scheme of Blackburn et al. (2011, Table 1). The latter scheme was also used to classify the invasive status of introduced psittacines in Puerto Rico.

Line 160 – “…. with at least 20 records were included in the subsequent analyses.” Eliminate “in this paper”.

Line 162 – Change to: “We assessed the current distribution of the introduced psittacines by surveying Ebird for geo-referenced records….”

Line 166 – Change to: “using methods of Falcón, el al. (2012) and…”

Line 167 – 170 – Change to: “Species distribution models were constructed with the Maximum Entropy Method (MaxEnt ver 3.4.1;…), which is a learning machine method that uses presence only data in combination with predictive variables to model a species’ geographic distribution”.

Lines 202-208 – Eliminate this introduction paragraph – too general.

Line 220 – Change to: “Many psittacines, including B. versicolurus roost….”

Line 233 – “observations on natural history information….” Change to “observations on food and nest resources (plant species).”

Lines 256 257 – Change to: “mark-recapture (Casagrande & Beissinger, 1997). These four methods produced 95% confidence intervals, which overlapped each other.”

Line 258 – “To test the effectiveness of this method…” Change to [?]: “To test the effectiveness of the mark-recapture method with the…”

Line 259 – Change to: “…, we conducted three counts spaced a week apart from September to October…”

Line 262 – Change to: “For estimates of population size and growth rates, we assumed that…”

Line 267 – Remove: “(these cannot be separated with this method)”

Line 299 – Change “On the other hand,…” to: At least 46 psittacid species are now present on the island (Fig 1, Table 1), of which 24% are only found in the pet trade, 48% have been observed in the wild (present), but not known to be breeding (established), and 28% are established (naturalized) and are known to have bred or are currently breeding.

Line 313 – “Of the species with population increase,…” Change increase to increases,..

Line 321 – “The green-checked amazon (….)…” Change to green-cheeked amazon.

Line 337 – Eliminate Puerto Rico from end of sentence (paper is on psittacid in PR, so save space).

Line 341 – change “were” to “where”

Line 342 – “Furthermore, the only other historic record are…” Change word “record” to records

Line 345 – “especially to coastal and heavily (human) populated zones,…” change to: especially in coastal and urban/suburban areas, but they….

Line 347 – Change to: To predict potential distribution of psittacine species whose populations are increasing, we obtained a total of 106,493 occurrence records from their native…

Line 357 - B. versicolurus – I suggest be consistent in use of scientific and common names. I suggest give scientific name with common name together when first used in the manuscript, then use common names afterwards. – In this paragraph you start with white-winged parakeets and then move to the scientific name and change back and forth – this gets confusing.

Line 367 – “Ecology of Brotogeris versicolurus in PR” – change to common name and use common name hereafter.

Line 385 – Eliminate – information on feather mites – not relevant to your manuscript.

Line 399 – “was estimated in 604 for San Germán …” Change to: “was estimated to be 604 for San Germán…”

Lines 405 – 410 –Repetitive, reduce length.

Lines 414- 434 – Repetitive – authors have repeatedly made the point that exotic psittacines on the island are a result of the pet trade. No need to repeat it. This paragraph should be eliminated – examples here are anecdotal – masked lovebird example could go in a table.

Lines 435 – 465 - This paragraph as the previous paragraphs is just a collection of anecdotes on the status of introductions – these anecdotes could perhaps go into an appendix, which would summarize, species by species the history and status of each introduced species. Discussion should summarize and synthesize patterns related to success or failure of exotic establishments.

Lines 466 – 468 – First paragraph repeats earlier information that exotics most common along coastal plan and in urban areas.

473 – 474 – Repetitive statement that white-winged and monk parakeets are most sighted and widespread species on island.

Line 478 – “…., and species more commonly found in captivity.” – change “more” to “most”.

Line 489 - This paragraph summarizes potential negative effects of introduced psittacines as found elsewhere, but is there any evidence for these negative effects in PR?

Line 505 – “before native birds have access to them (because native species only eat them when they are ripe – I am skeptical of this statement – many native species will take unripe fruit including bullfinches and spindalis and other tanagers and finches.

Line 506 – Correct this citation: Snyder, Wieley & Kepler 2007…..” Change “Wieley” to Wiley (correct on line 509 also).

Line 517 – Paragraph – “Another aspect to consider is that the three species of endemic parrots in Puerto Rico were once much more common and used to occupy many of the areas and habitats now occupied by exotic species” - The current habitats occupied by the exotic species is likely very different from the habitat occupied by the endemic psittacines – the exotics are using habitats with many exotic plant species (e.g., “novel” communities) including second growth forests/woodlands and suburban and urban areas, which were not occupied by the endemics. Thus as with the exotic finches (see Raffaele, A.H. 1989. The ecology of native and introduced granivorous birds in Puerto Rico. In Biogeography in the West Indies: past, present, and future: 541–566. Woods, C.A. (Ed.). Gainesville, FL: Sandhill Crane Press), the exotic psittacines are occupying novel habitats which were not available to the extinct endemics – in other words the exotic psittacines are occupying vacant niches, which may not have been available for occupancy by the endemics.

Lin 520 – Change “damages” to word “damage”.

Line 525 – “Parrots are known to incur in such functions…” Change to: Some parrots are known to be both seed predators and dispersers…” Nothing or little is known about the PR parrakeet’s and parrot’s role as seed predators or dispersers, so this seems highly speculative.

Line 531 – “and they often predate on them in the wild” Change to: and they often destroy seeds in the wild.

Line 533 – “But they also predate seeds of the pink trumpet…” Change to: “But they also depredate seeds of native species such as pink trumpet…..”

Line 535 – Change “predate” to word “depredate.”

Line 547 – “However, rarely the number of individuals trapped was reported”. Change to: However, the number of individuals trapped was rarely reported.”

Line” 553 – 560 – Paragraph here is about recommendations for sale and reporting of psittacine sales – is not relevant to this paper. Eliminate

Lines 561- 568 - “If control and management is necessary and/desireable…” Paragraph focus is not relevant to paper. Eliminate – recommendations are found elsewhere in literature and not unique to PR.

Line 570 – Ecology of Brotogeris versicolurus in PR – Use common name in title, to be consistent.

Line 577 –583 --Eliminate the mite discussion – perhaps publish in a separate note, but not relevant here. Eliminate this paragraph as natural history notes on mites are irrelevant to your study, or you do not make a connection to the study. Publish as a separate note.

Line 589 – This paragraph is highly speculative – can you be certain you’ve found all of the roost sites?

Line 602-603 – Change to: “Additionally, as the size of a species’ population increases, so does the geographic distribution (Veit 1997), so we can expect the population result in spread….”

Line 604- “(although this is not always the case,”). Shorten to “(but not always, …”).

Line 605-607 – Eliminate sentence about Yellow-headeded Blackbird details – represent unnecessary details – you can cite Veit (1997) in earlier sentence as suggested.

Line 608 – “….have a proportional increase of long dispersers…: Change to: “…have a proportional increase in long dispersing individuals (vagrants)….”

Line 608 – Change word “incrementing” to “increasing”.

Line 618 - “populations suggested that they were they only existing…” Change to: “populations suggested that they were the only existing roosting…”

Line 622 – Change “variations” to word “variation”. (singular not plural).

Line 633 – “Puerto Rico and parks…” Change to: “highly abundant in Puerto Rican forests and parks.”

Line 638 – Change word “exacerbating releases” to: increasing the parakeet numbers in the wild.

Line 645 – Eliminate word “indeed”.

Line 655 – Eliminate paragraph on PR parrot as your research findings do not address this issue.

Line 661 – This paragraph on hurricane impacts does not belong in the conclusions as your research findings are unrelated to hurricanes – Conclusions should focus on your research findings, and hurricane discussion here is irrelevant. Eliminate entirely or move elsewhere.

Line 676 – This paragraph on studying endangered exotics does not belong in the conclusions section – perhaps move to Discussion section.

Line 698 – Change “Wieley” to “Wiley”.

·

Basic reporting

A few small issues with the English language.

For main comments see below.

Experimental design

For main comments see below.

The information on the roost counts was not presented in sufficient detail.

Validity of the findings

See comments below

Additional comments

In the paper “From the cage to the wild: Introductions of Psittaciformes to Puerto Rico with emphasis on the invasive ecology of the white-winged parakeet” the authors have made a useful compilation of information on the introduced parrots of Puerto Rico. They have also gone more in depth in to the status of the White-winged Parakeet. The manuscript is overall relatively easy to follow, but it has some rather major issues that will need to be addressed.

Starting with the title, it is clear that the paper has two main themes. Introduced parrots in general and White-winged Parakeets in specific. Unfortunately, the paper reads like two papers that have been put together in the same manuscript. I think that the authors need to separate out the two topics in to different papers and submit each individually to either the same or different journals. By restructuring in this way, the authors will be able to devote more attention to each manuscript and make each stronger on its own.

The authors use a Maxent analysis using a combination of locations from the native range and introduced range to predict potential distribution on the island. Most observations of the introduced birds in Puerto Rico fall within the predicted Maxent ranges, but not all. It is unclear to me why the authors chose to do a single Maxent map using a mixture of locations from native and introduced ranges. The authors should provide better justification for this. Or ideally, do two Maxent maps one using locations from the native range and another using locations from Puerto Rico to determine how they differ. If the overview of introduced species is separated off in to a separate paper, it will provide enough space to include an analysis like this and its discussion.

However, there is a single much larger issue with the spatial / habitat analysis in this paper. The authors have not addressed the issue of urban versus native habitat use by the introduced parrots. One classic hallmark of introduced parrots is that they often use preferentially (or sometimes exclusively) anthropogenic habitats. From looking at the maps generated, many of the clusters of points signifying high use by introduced parrots overlap with the population centers on the island. This is an issue that must be addressed in the spatial analyses through the use of some sort of layer that indicates urbanization or the like.

However, there is another reality that could be adding to this finding. The eBird locations may be preferentially clustered in areas with more people. Are there well known parks with high numbers of eBird checklists that do not report introduced parrots? The authors need to discuss both the quantitative and anecdotal/qualitative information about urban versus natural habitat use by the introduced parrots.

The authors also need to discuss the use of eBird data in general. Sometimes eBirders use county or city locations when entering their data. As a result, the actual point location is not very accurate. How did the authors filter or check their data for this? While eBird is a wonderful resource, the authors need to discuss points from the literature about possible strong and weak points of these data and how that may have impacted their findings.

With regards to the focus on the White-winged Parakeets, the authors should go in to more detail about the censuses they did. In general this part of the paper seemed “rushed.” There was an overall lack of detail, development, and discussion. Again, if this is broken in to its own paper, the authors will have the opportunity to develop this information fully.

---

## Round 0.2 · Major Revisions

The prior reviewers were not available to re-review and so I brought in a new reviewer. Although the referee does not question the development of the work itself, they mention that a major edit of the English language is necessary. Please also follow the comments to reduce the discussion.

I hope you can send it soon to a review with a colleague and that we will receive the corrected paper again. Thank you for choosing PeerJ

·

Basic reporting

Although I am not a native English speaker, I find that it there are many errors of language and syntax, making the readability of the manuscript not fluent. A severe language polishing is recommended.

Experimental design

OK

Validity of the findings

OK

Additional comments

Line 42: delete “where they are introduced”
Line 68: change “Among the species commonly sold as exotics are Psittaciformes” with “Psittaciformes are among the exotics species most commonly sold as pets”
Line 76. This is true, but you have to say that most “unsuccessful” introductions have not been documented. See, for instance: Mori E., Monaco A., Sposimo P., Genovesi P. (2014). Low establishment success of alien non-passerine birds in a Central Italy wetland. Italian Journal of Zoology, 81: 593-598.
Lines 110-114: I suggest you to take a look at this: https://www.inaturalist.org/projects/alien-parrots-observatory
Line 212. Delete the dot after “Psittacara”
Line 272. Change “sought to assess” with “assessed”.
The Discussion chapter is too long and unbalanced with respect to the rest of the MS. I suggest you to summarise it and to reduce it at least by 40%.

Emiliano Mori

---

## Round 0.3 · accepted · Accept

I believe that you have taken into account the comments of the reviewers. I think your work is ready to be published in PeerJ

#